# Sensitive visualization of SARS-CoV-2 RNA with CoronaFISH

Elena Rensen[1,]*, Stefano Pietropaoli[2,]*, Florian Mueller[1,]*, Christian Weber[1,]*, Sylvie Souquere[3,†], Sina Sommer[2,†], Pierre Isnard[4,5], Marion Rabant[4,5], Jean-Baptiste Gibier[6], Fabiola Terzi[4], Etienne Simon-Loriere[7], Marie-Anne Rameix-Welti[8,9], Gérard Pierron[10], Giovanna Barba-Spaeth[2], Christophe Zimmer[1]

**The current COVID-19 pandemic is caused by the severe acute respiratory syndrome coronavirus 2 (SARS-CoV-2). The positive-sense single-stranded RNA virus contains a single linear RNA segment that serves as a template for transcription and replication, leading to the synthesis of positive and negative-stranded viral RNA (vRNA) in infected cells. Tools to visualize vRNA directly in infected cells are critical to analyze the viral replication cycle, screen for therapeutic molecules, or study infections in human tissue. Here, we report the design, validation, and initial application of FISH probes to visualize positive or negative RNA of SARS-CoV-2 (CoronaFISH). We demonstrate sensitive visualization of vRNA in African green monkey and several human cell lines, in patient samples and human tissue. We further demonstrate the adaptation of CoronaFISH probes to electron microscopy. We provide all required oligonucleotide sequences, source code to design the probes, and a detailed protocol. We hope that CoronaFISH will complement existing techniques for research on SARS-CoV-2 biology and COVID-19 pathophysiology, drug screening, and diagnostics.**

## Introduction

Coronavirus disease (COVID-19) emerged by the end of 2019 in Wuhan, China, and led to more than 240 million infections and over 5 million deaths as of 1 November 2021 (Johns Hopkins University Dashboard). Its causative agent, severe acute respiratory syndrome coronavirus 2 (SARS-CoV-2), is an enveloped, positive-sense, single-stranded RNA virus. Upon infection, viral replication occurs in the host cell's cytoplasm, which is massively reorganized (V'kovski et al,

2020). The genomic positive-strand viral RNA (vRNA) serves as a template for transcription and replication. The virus synthesizes its own RNA-dependent RNA polymerase (RdRP) to generate negative-sense RNA replication intermediates. This negative strand acts as template for replication of new full-length positive-stranded RNA genomes and for transcription of several smaller, subgenomic positive-stranded RNAs (sgRNAs). These sgRNAs are then used to synthesize all other viral proteins in spatially confined replication complexes. Mature virions are exocytosed and released from the infected host cell. Despite recent progress, many aspects of the SARS-CoV-2 viral replication cycle, including the subcellular location of vRNA synthesis, are still not fully understood and under active investigation (V'kovski et al, 2020).

Several established techniques allow studying SARS-CoV-2 and its interaction with its host. Immunofluorescence (IF) permits the visualization of viral and host proteins in the spatial context of a single cell. However, the development of specific antibodies against novel viruses is time- and cost-intensive, especially if specificity over other closely related viruses is required. Furthermore, the presence in cells of structural viral proteins, such as the Spike protein, does not necessarily imply active viral replication (Cheung et al, 2005; Tang et al, 2020) and their subcellular localization may not reflect that of the vRNA strands. Other molecular methods, such as RT-PCR, provide an accurate, quantitative readout of viral load and replication dynamics, but are bulk measurements over large cell populations that mask variability between cells and provide no information about the subcellular localization of the virus. RNA-seq permits a complete view of the transcriptome of both the host and virus, including in single cells, albeit again without spatial information (Bost et al, 2020; Kim et al, 2020).

Unlike immunostaining, PCR, or sequencing methods, FISH offers the ability to directly and specifically visualize vRNA in single cells

[1]Institut Pasteur, Université de Paris, CNRS UMR 3691, Imaging and Modeling Unit, Paris, France [2]Institut Pasteur, Université de Paris, CNRS UMR 3569, Unité de Virologie Structurale, Paris, France [3]Gustave Roussy, AMMICA UMS-3655, Villejuif, France [4]Université de Paris, INSERM U1151, CNRS UMR 8253, Institut Necker Enfants Malades, Département "Croissance et Signalisation," Paris, France [5]Service d'Anatomo-Pathologie, AP-HP, Hôpital Necker Enfants Malades, AP-HP Centre, Paris, France [6]Service d'Anatomo-Pathologie, Centre de Biologie Pathologie, CHU Lille, Lille, France [7]Institut Pasteur, Université de Paris, G5 Evolutionary Genomics of RNA Viruses, Paris, France [8]Université Paris-Saclay, INSERM, Université de Versailles St. Quentin, UMR 1173 (2I), Montigny-le-Bretonneux, France [9]AP-HP, Université Paris Saclay, Hôpital Ambroise Paré, Laboratoire de Microbiologie, Boulogne-Billancourt, France [10]Gustave Roussy, CNRS UMR 9196, Villejuif, France

Correspondence: czimmer@pasteur.fr; florian.muller@pasteur.fr; Gerard.PIERRON@gustaveroussy.fr; giovanna.barba-spaeth@pasteur.fr
Stefano Pietropaoli's present address is Catalent Pharma Solutions, Strada Provinciale 12 Casilina, 41, Frosinone, Italy
*Elena Rensen, Stefano Pietropaoli, Florian Mueller, and Christian Weber contributed equally to this work.
†Sylvie Souquere and Sina Sommer contributed equally to this work.

(Raj et al, 2008; Itzkovitz & van Oudenaarden, 2011; Mueller et al, 2013). In RNA-FISH, single RNA molecules are typically targeted with 10–50 fluorescently labeled probes consisting of short (20–30 nucleotides), custom synthesized oligonucleotides with sequences designed in silico. Individual RNAs are subsequently visible as bright, usually diffraction-limited spots under a microscope, and can be analyzed with appropriate image analysis methods (Raj et al, 2008; Mueller et al, 2013). Single molecule FISH (smFISH) has been used in biological samples ranging from single-cell organisms such as bacteria and yeast to whole tissue sections and organs (Trcek et al, 2012; Skinner et al, 2013; Chen et al, 2016; Wu et al, 2018; Wang, 2019). We recently introduced single molecule inexpensive FISH (smiFISH) (Tsanov et al, 2016), an inexpensive variation of smFISH that is ideally suited to visualize RNA viruses and study their subcellular localization and kinetics in host cells (King et al, 2018; Rensen et al, 2021).

In this study, we designed and validated smiFISH probes against the positive and negative RNA strands of SARS-CoV-2 (CoronaFISH). We demonstrate highly specific viral detection in cell culture, in human tissue samples and in patient isolates. We further demonstrate the flexibility of these probes by adapting them for electron microscopy in situ hybridization (EM-ISH). CoronaFISH provides a flexible, cost-efficient and versatile platform for studying SARS-CoV-2 replication at the level of single cells in culture or in tissue, and can potentially be used for drug screening and diagnosis.

# Results

## Design of probes specific for SARS-CoV-2

Our RNA-FISH approach uses two types of bioinformatically designed DNA oligonucleotides (oligos) (Tsanov et al, 2016): (i) unlabeled primary oligos consisting of two parts: a specific sequence complementary to a selected subregion of the target RNA and a readout sequence that is identical among all primary oligos (FLAP sequence), (ii) a fluorescently labeled secondary oligo complementary to the FLAP sequence, allowing visualization by light microscopy. These oligos are hybridized to each other in vitro before their use for cellular imaging (Fig 1A).

A cell infected by SARS-CoV-2 can contain the incoming positive strand, the negative-strand replication intermediate, as well as replicated full-length and subgenomic positive-strand RNA molecules (Fig 1B). We designed two sets of 96 probes, one against the positive strand and one against the negative strand of SARS-CoV-2 (Fig 1C). For more details on the probe design workflow, see the Materials and Methods section and the source code (https://github.com/muellerflorian/corona-fish). In brief, we identified an initial list of more than 600 potential probe sequences with our previously described method Oligostan (Tsanov et al, 2016). We then further screened these probes to be robust to known genomic variations of SARS-CoV-2 (as of April 2020), while removing probes with affinity to other known β-coronaviruses or viruses frequently causing similar respiratory diseases in humans such as Influenza. Last, we removed probes overlapping with the transcriptome of several relevant host

organisms (human, mouse, African green monkey, hamster, and ferret). To establish the final list of 96 probes (Fig 1C), we chose probes targeting regions with the highest sequencing coverage. The complete list of probe sequences is provided in Table S1.

## Visualization of SARS-CoV-2 in Vero cells with CoronaFISH

To test our probes, we first used Vero cells (African green monkey), which are highly permissive for SARS-CoV-2 replication (Takayama, 2020). We processed the samples for FISH following the protocol detailed in Supplemental Data 1 and imaged cells infected with SARS-CoV-2 (Wuhan/IDF strain) 18 h post infection (p.i.) with a MOI of 0.2, as well as uninfected control cells. The positive-strand and negative-strand probe sets were both labeled with the fluorophore Cy3 and imaged separately in distinct experiments.

In uninfected samples, cells displayed only weak and diffuse fluorescent signal when labeled with either probe set, consistent with unspecific background labeling common in RNA-FISH (Tsanov et al, 2016), and occasional bright spots could be observed, mostly located outside the cells, presumably because of unspecific probe aggregation (Figs 1D and E and S1A). By contrast, in infected samples, a large proportion of cells showed very strong and localized signal in large regions of the cytoplasm (Figs 1D and E and S1A). Quantification of the fluorescence intensities per cell indicated that 26% of cells in the infected sample labeled for the positive strand (n = 242) had intensities exceeding a threshold that excluded >99% of uninfected cells (n = 83) (Fig 1E). The fluorescence intensity in these cells was on average 23 times higher than this threshold (s.d. 27) (Fig 1E). For the negative strands, we counted 5% of Vero cells with intensities above the similarly defined threshold in infected samples (n = 307), versus <1% for uninfected cells (n = 103), with on average 2.5 times higher intensities (s.d. 2.2).

The fluorescent signal for the positive-strand was remarkably strong compared to the uninfected control images (Fig 1E). This is consistent with an extremely high per-cell viral yield, which has previously been reported for Vero cells (Ogando et al, 2020). The RNA signal was perinuclear and restricted to the cytoplasm, consistent with cytoplasmic replication, as previously reported for other Coronaviridae and recently for SARS-CoV-2 (Stertz et al, 2007; Fung & Liu, 2019; Klein et al, 2020; Snijder et al, 2020). Interestingly, we observed bright foci of different sizes and intensities, some of which displayed hollow structures reminiscent of the replication organelles (ROs) or double-membrane vesicle (DMV) structures described previously (Klein et al, 2020; Ogando et al, 2020). The signal for the negative strand was substantially dimmer than for the positive-strand (Fig 1E). This is consistent with previous reports that the replication intermediate negative strand is less abundant (Wolff et al, 2020), although we cannot exclude that this is due to reduced accessibility of the negative strand to the probe set. The negative-strand RNA likewise forms clusters of different sizes and intensities in the proximity of the nucleus corresponding to the location of DMVs (Figs 1D and S1A).

To further validate the specificity of CoronaFISH, we combined positive-strand imaging with immunostaining using J2 antibody. J2 specifically detects dsRNA and has previously been used to identify SARS-CoV-2 ROs (Cortese et al, 2020). In dual-color imaging experiments at 6 h p.i., we observed both the

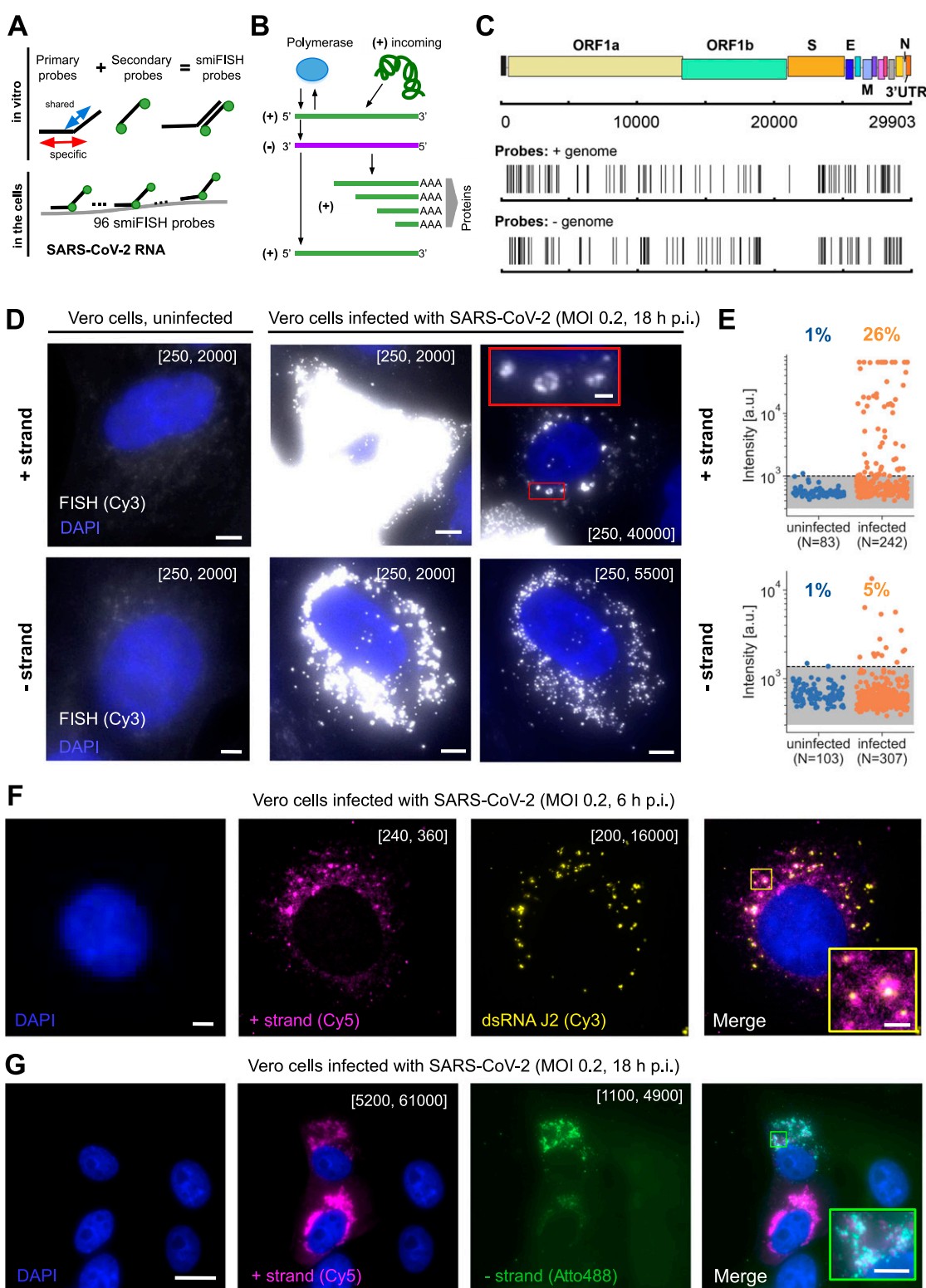

**Figure 1. Visualizing SARS-CoV-2 RNA with CoronaFISH.**
**(A)** Principle of CoronaFISH. 96 primary probes are prehybridized in vitro with dye-carrying secondary probes via the shared FLAP sequence. Resulting duplexes are subsequently hybridized in cells to target the SARS-CoV-2 positive or negative RNA. **(B)** Replication cycle of SARS-CoV-2. Incoming, genomic positive-strand RNA (green) is used to produce viral polymerase. Polymerase produces a negative-strand replication intermediate (violet), which serves as a template for synthesis of full-length positive-strand and shorter subgenomic RNAs (green). The latter are used to produce other viral proteins. **(C)** Genome of SARS-CoV-2 with indicated probe positions targeting the positive and negative strand. **(D)** CoronaFISH images of uninfected and infected Vero cells with either the positive (top) or negative (bottom) strands

positive-strand SARS-CoV-2 RNA with CoronaFISH, and the dsRNA with J2 immunostaining (Fig 1F), whereas this was not the case in uninfected control cells (Fig S1B). Because CoronaFISH labels the positive RNA strands, whereas J2 labels the dsRNA, the two signals are expected to coincide in regions of viral replication. Although the quality of the CoronaFISH image was lower than in previous experiments, likely due to the different sample preparation for immunolabeling, we could still clearly observe partial colocalization of positive strand RNA with J2 antibody in bright foci, suggesting that these are sites of active viral replication (Fig 1F). These data thus provide additional validation of SARS-CoV-2 detection by CoronaFISH.

We next wanted to demonstrate the ability of CoronaFISH to simultaneously visualize positive and negative SARS-CoV-2 RNA with dual-color imaging (Tsanov et al, 2016). For this purpose, we used different FLAP sequences on the primary probes set (FLAP-X for positive-strand, and FLAP-Y for negative-strand probes), and labeled them with spectrally different fluorophores (Cy5 and Atto488, respectively). We then imaged infected Vero cells at 18 h p.i. These images clearly show the presence of both positive and negative RNA strands in the same cells and the same subcellular regions, although the relative abundance of both strands appeared to vary from cell to cell and colocalization was only partial (Figs 1G and S1C). By allowing visualization of positive and negative RNA together, CoronaFISH opens the door to analyzing the intracellular organization and dynamics of viral transcription and replication.

Our data thus show that CoronaFISH probes can sensitively and specifically detect the positive and negative strands of SARS-CoV-2 in infected Vero cells.

## Visualization of SARS-CoV-2 for varying levels of infection

To further investigate the sensitivity of CoronaFISH, we infected Vero E6 cells with SARS-CoV-2 (D614G strain) at MOIs spanning two orders of magnitude (0.1, 0.01, and 0.001) and imaged cells 24 h p.i. Quantification of the virus released in the supernatant of infected cells by a focus forming assay (FFA) showed viral production increasing by roughly 10-fold for each 10-fold increase in MOI (Figs 2A and S2A).

When imaging these cells with CoronaFISH, we observed a clear increase in the number of infected cells with increasing MOI (Fig 2B). This trend was confirmed by automated quantitative image analysis on three replicates (Fig 2C and D). Note that among infected cells, the intensity values computed from the images varied over a large range, consistent with strong differences in viral load (Figs 2C and S2C). These data are consistent with earlier studies showing that viral infection is driven by a subpopulation of permissive cells in which the virus can replicate at high levels (Lee et al, 2021 *Preprint*).

Our data using varying MOIs thus illustrate the sensitivity of CoronaFISH and how it can be used to distinguish cell populations with different viral loads—an observation that would be averaged

out by population measurements. Unlike bulk assays, CoronaFISH allows for spatiotemporal analysis of SARS-CoV-2 replication and spread.

## Applicability of CoronaFISH to drug screening

We next wanted to further probe the sensitivity of CoronaFISH and its applicability to testing pharmacological treatments against SARS-CoV-2. To this end, we treated SARS-CoV-2 (D614G strain) infected Vero E6 cells with Remdesivir, an adenosine nucleoside triphosphate analog that reduces viral replication in vitro by inhibiting the RdRP (Gordon et al, 2020).

We first established an inhibition curve using FFA by infecting Vero E6 cells with SARS-CoV-2 D614G strain with an MOI of 0.1, and treating cells with Remdesivir concentrations ranging from 0.1 to 100 $\mu$M (Fig S2B and D). This curve yielded a calculated IC50 concentration of 1.12 $\mu$M. To better probe the sensitivity of CoronaFISH, we then turned to a very low MOI of 0.001 and analyzed in more detail Remdesivir concentrations ranging from 0 to 10 $\mu$M. The virus titer in the supernatant of treated cells decreased with increasing Remdesivir concentration and dropped sharply between 0.86 and 3 $\mu$M, consistent with the above IC50 value (Fig 2E).

We then used CoronaFISH to image the positive SARS-CoV-2 RNA strand in infected cells for the entire range of Remdesivir concentrations, in two replicates. Visual inspection of the images revealed a strong decrease in the percentage of infected cells with increasing Remdesivir concentrations (Fig 2F). Quantitative analysis of the images confirmed this (Fig 2G and H), and interestingly, also revealed a reduction in average intensity for increasing Remdesivir concentrations (Figs 2G and S2E). Of note, even at a high Remdesivir concentration of 10 $\mu$M, we could find a small fraction of isolated cells with a weak but clear CoronaFISH signal (Fig 2F and H).

These data not only confirm the sensitivity of CoronaFISH but also show that it can be used to test molecules for their ability to inhibit SARS-CoV-2 replication at the single cell level.

## Detection of SARS-CoV-2 RNA in human cell lines

We next tested the CoronaFISH probes in several human cell lines known to be susceptible to SARS-CoV-2 (Takayama, 2020): Caco-2 (human intestinal epithelial) cells, Huh7 (hepatocyte-derived carcinoma) cells, and Calu-3 (human lung cancer) cells. Each cell line was infected with SARS-CoV-2 (Wuhan/IDF strain) at an MOI of 0.2 and fixed at 36 h p.i. Titration on the supernatants of the infected cells revealed vastly different replication efficiencies as tested by FFA (Table 1). Caco-2 and Huh7 cells released rather low virus levels, in the same order of magnitude as Vero cells (2–5 × $10^3$ FFU/ml), whereas Calu-3 produced two orders of magnitude higher levels of infectious virus (2 × $10^5$ FFU/ml).

---

detected with Cy3-labeled probes (white). Nuclei are labeled with DAPI (blue). Shown are zoom-ins on individual cells. Full-size images are shown in Fig S1A. First column shows uninfected control cells, second and third columns show infected cells with different intensity scaling as indicated in brackets (the first and second values in brackets indicate the pixel values corresponding to the lowest and brightest intensities in the displayed image, respectively). Scale bars: 5 $\mu$m. Scale bar in red inset: 1 $\mu$m. **(E)** Quantification of signal intensities in individual cells. Dashed line is the 99% quantile estimated from uninfected samples. **(F)** CoronaFISH image of the positive strand combined with immunofluorescence detection of dsRNA with the J2 antibody. Scale bar in full image: 2 $\mu$m, in inset: 0.5 $\mu$m. **(G)** Dual-color imaging of positive and negative strands. Scale bar in full image: 10 $\mu$m, in inset: 2 $\mu$m. Intensity scalings in panels (F) and (G) are shown as described for panel (D).

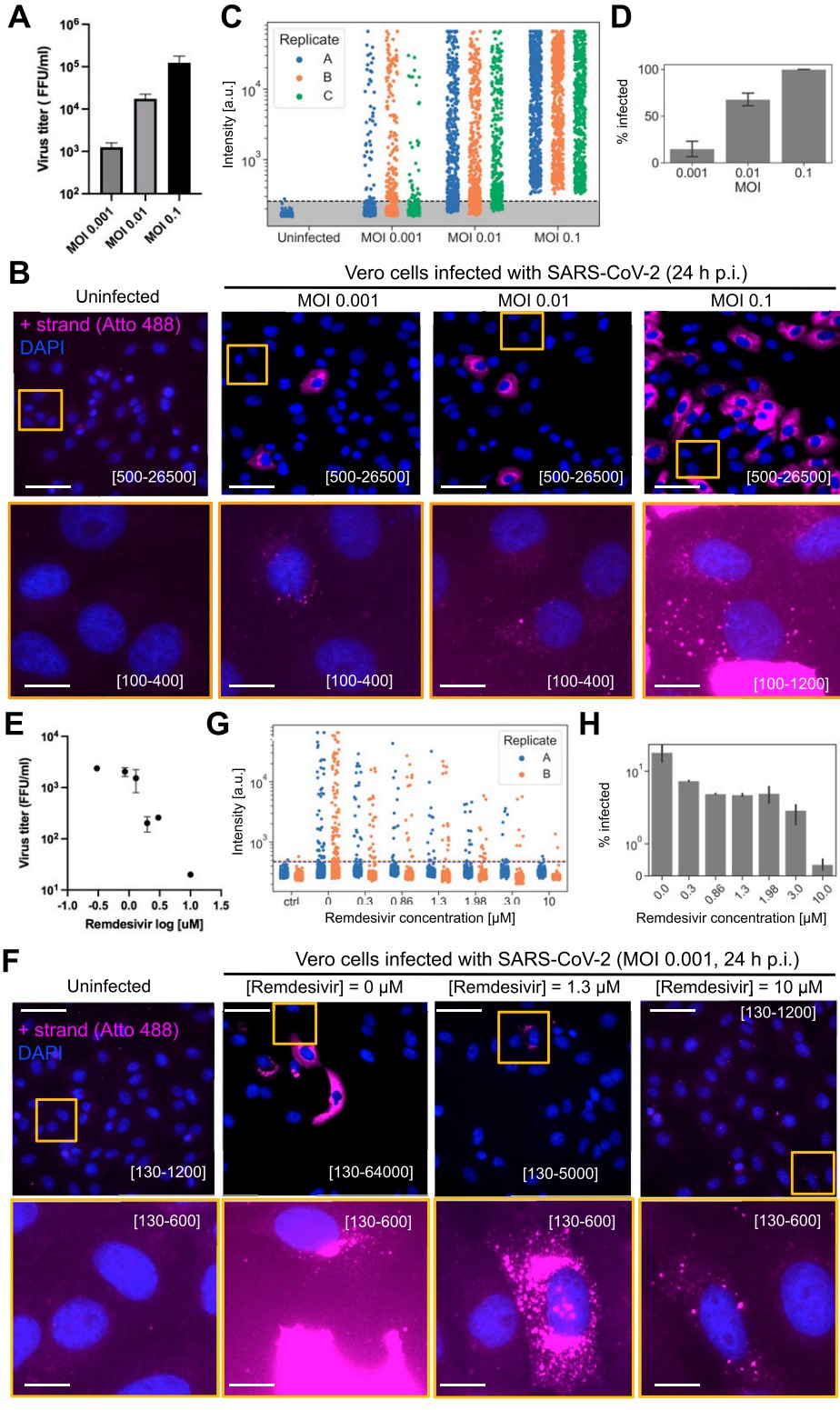

Figure 2. Visualizing SARS-CoV-2 with different MOIs and pharmacological inhibition.
(A) Viral titers as function of MOI, as quantified by a focus forming assay (FFA). FFA measures the titer of infectious virus released 24 h p.i. from Vero E6 cells infected with SARS-CoV-2 D614G virus at MOIs 0.001, 0.01, and 0.1. (B) CoronaFISH images of uninfected and infected Vero E6 cells with different MOIs, at 24 h p.i. The positive SARS-CoV-2 RNA signal is shown in pink, and nuclei (DAPI) in blue. Shown are full-size images (upper row) and zoom-ins on individual cells (lower row). Intensity scalings are indicated in brackets as in Fig 1D. Scale bar in full image: 50 $\mu$m, in inset: 10 $\mu$m. (C) Quantification of RNA-FISH intensities in individual cells for three replicates (A, B, C). Dashed line is the 99.9% quantile estimated from uninfected samples. (D) Quantification of the percentage of infected cells. Shown are the mean and standard deviations of the three replicates. (E) Quantification of viral inhibition by Remdesivir using FFA. The plot shows the titers of infectious virus released 24 h p.i. from Vero E6 cells infected with SARS-CoV-2 D614G and treated with different doses of Remdesivir (0.86–10 $\mu$M). (F) Images of infected Vero E6 cells (MOI 0.001, 24 h p.i.) treated with different concentrations of Remdesivir. The positive SARS-CoV-2 RNA signal is shown in pink, and nuclei (DAPI) in blue. Shown are full-size images (upper row) and zoom-ins on individual cells (lower row). Intensity scalings are indicated in brackets as in Fig 1D. Scale bar in full image: 50 $\mu$m, in inset: 10 $\mu$m. (G) Quantification of signal intensities as function of Remdesivir concentration, for two replicates (A, B), displayed as in panel (C). (H) Percentage of infected cells computed from (G). Shown are the mean and standard deviations of the two replicates.

We then performed FISH against the positive RNA strand. As for the Vero cells above, the uninfected controls of all human cell types showed only a weak background signal, whereas a strong signal could be detected in the infected cells (Fig 3A–C). However, depending on the cell type, the number of infected cells, as well as the amount and cellular localization of vRNA detected by CoronaFISH were very different, consistent with different replication dynamics of SARS-CoV-2 in these cell lines. In Caco-2 cells, only a

**Table 1. Titration on the supernatant of infected mammalian cells.**

| Cell line | Viral titer (FFU/ml) |
|---|---|
| Caco-2 | $2 \times 10^3$ |
| Huh7 | $4 \times 10^3$ |
| Calu-3 | $2 \times 10^5$ |
| Vero | $5 \times 10^3$ |

minority of cells appeared infected (19% of n = 1,752 cells above intensity threshold defined from n = 229 uninfected cells as above), displaying rather low-intensity values, indicating a low abundance of positive-strand vRNA (6.2-fold above threshold, s.d. 6.8). Huh7 cells were more permissive for viral infection, as indicated by a higher percentage of cells with intensities above threshold (74% of

n = 546 above threshold defined on n = 246 uninfected cells) and higher RNA signal intensity (11-fold above threshold, s.d. 10). Last, all Calu-3 cells were infected (100% of n = 773 cells above a threshold defined on n = 479 uninfected cells) and the signal intensity was higher than in Caco-2 and Huh7 cells (28-fold above threshold, s.d. 14). These data show that CoronaFISH probes can also be used in cell lines of human origin with similar detection performance as in Vero cells.

## Detection of SARS-CoV-2 RNA in human tissue

Next, we aimed to test if CoronaFISH can be used to detect SARS-CoV-2 RNA in lung tissue samples derived from patients. The major histopathological finding of the pulmonary system of post mortem COVID-19 patients with acute respiratory distress syndrome (ARDS)

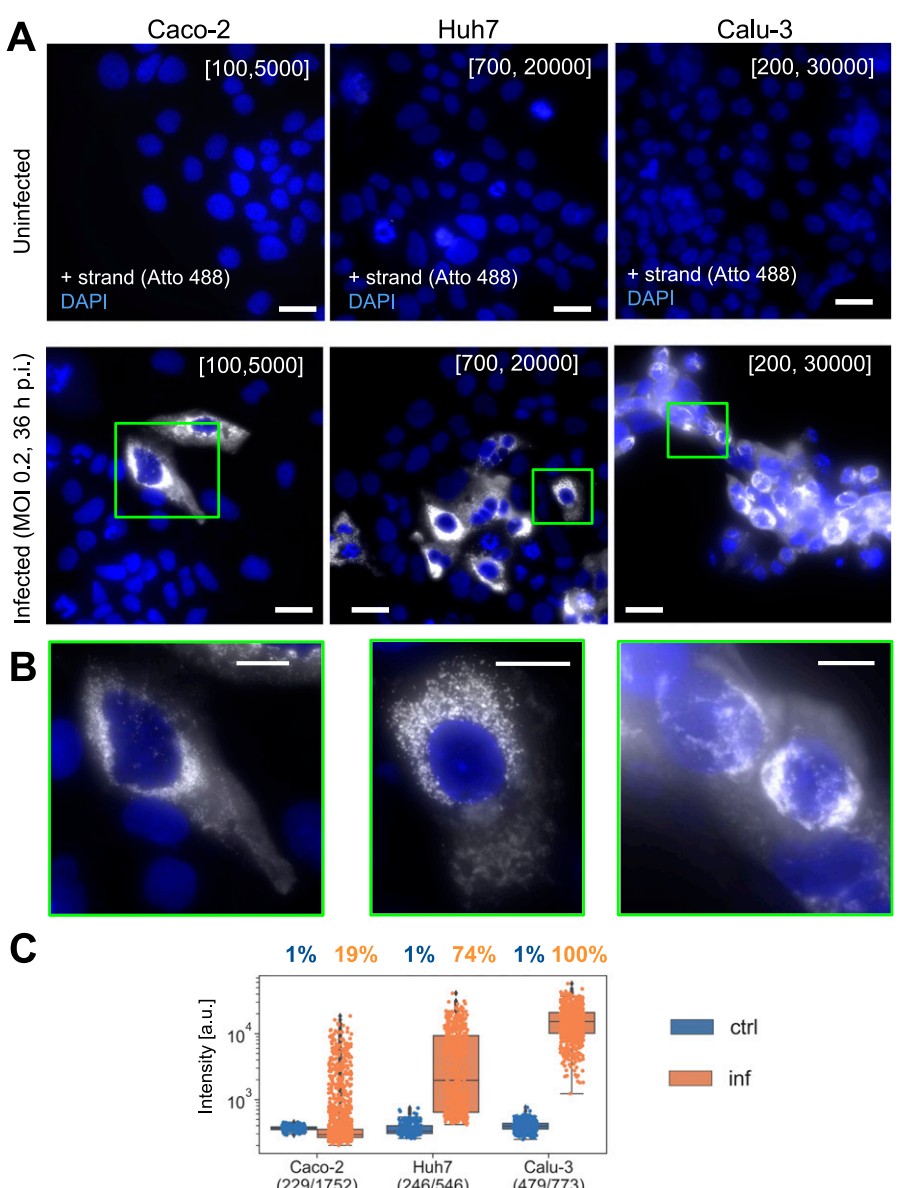

**Figure 3. CoronaFISH in human cell lines.**
**(A, B)** Visualization of positive-strand SARS-CoV-2 RNA in three different human cell lines (Caco-2, Huh7, and Calu-3). **(A)** Full field of views, scale bars 30 μm. Top row shows uninfected cells, bottom row shows cells infected by SARS-CoV-2 with MOI of 0.2 at 36 h p.i. Intensity scalings are indicated as described in Fig 1D. **(B)** Zoom-ins on indicated green rectangles, scale bars 10 μm. **(C)** Quantification of signal intensities of individual cells. Blue, uninfected control cells; orange, infected cells. Numbers in brackets indicate control cells and infected cells, respectively.

is diffuse alveolar damage in the acute or organizing phases. Lung tissue examination mainly shows evidence of intra-alveolar hemorrhage and edema with fibrin and hyaline membranes developed on alveolar walls at the acute phase and proliferative and fibrotic lesions in the alveolar septal walls at the organizing phase (Bradley et al, 2020; Hanley et al, 2020). However, these lesions are common to multiple forms of ARDS and not specific to COVID-19 and do not shed light on the underlying etiology. CoronaFISH therefore has the potential to specifically detect SARS-CoV-2 infection and characterize viral tropism within the tissue.

As a negative control, we imaged a tissue section sample obtained from a deceased adult patient with diffuse alveolar damage from ARDS before the COVID-19 pandemic. Histological analysis showed diffuse alveolar damage with an important alveolar hemorrhage, an intra-alveolar and interstitial edema with polymorphic inflammatory infiltrate and the presence of early hyaline membrane adjacent to alveolar walls (Fig S3A). When staining this sample with the positive-strand CoronaFISH probes, no strong fluorescent signal was detected, despite the presence of alveolar damage comparable to patients suffering from COVID-19 (Fig S3A–D).

As a positive control, we used samples from a COVID-19 patient who died 3 d after admission to the intensive care unit. Histological analysis showed diffuse alveolar damage at the organizing phase with intra-alveolar hyaline membranes and fibrin together with interstitial fibrotic lesions with polymorphic inflammatory cell infiltrate of alveolar walls (Fig S3D). Whereas histology by itself is not sufficient to diagnose lung tissue infection, CoronaFISH against the positive strand of SARS-CoV-2 RNA revealed infected cells with large cytoplasmic RNA aggregates (Fig 4A), illustrating that viral presence can also be detected in tissue sections. Because of the extensive destruction of tissue architecture, determining the affected area of the parenchyma is challenging. However, the cell types (e.g., macrophages or type 2 pneumocytes) infected by SARS-CoV-2 could potentially be revealed by combining CoronaFISH with immunostaining.

### Detection of SARS-CoV-2 RNA in nasal swabs

Motivated by the previous results, we next attempted to detect viral presence in human samples used for COVID-19 diagnostics. Nasal swabs were collected from patients with respiratory symptoms as

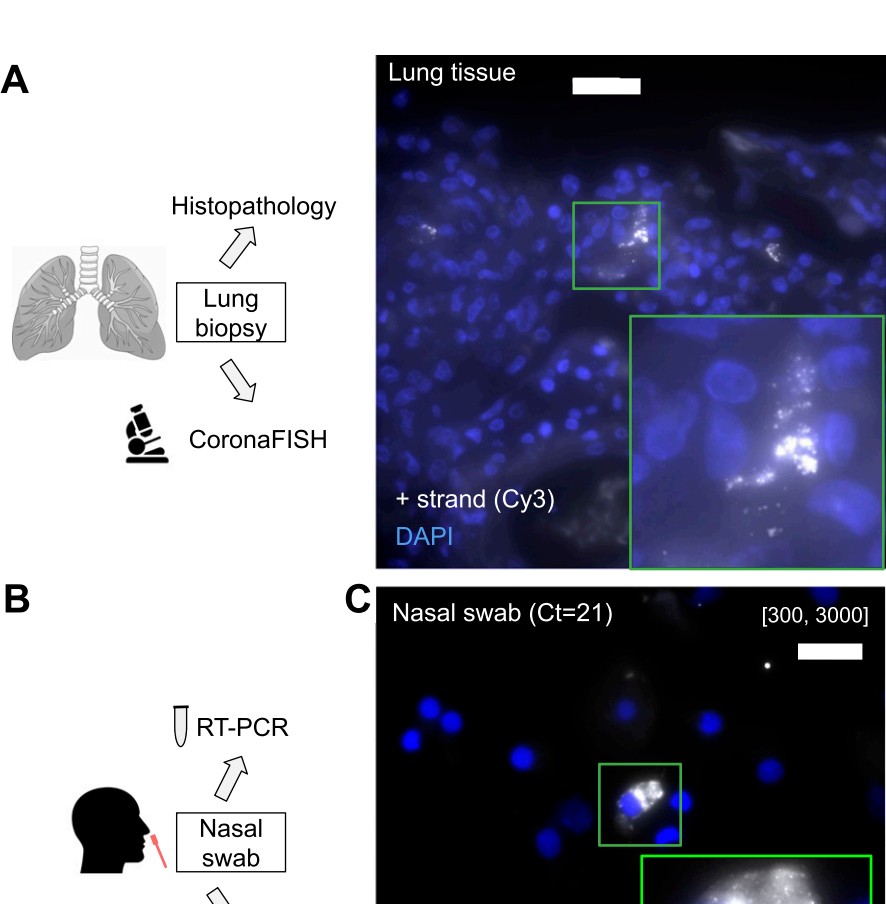

**Figure 4. CoronaFISH in human tissue and nasal swabs.**
**(A)** Visualization of positive-strand SARS-CoV-2 RNA in human lung tissue of a COVID-19 patient. Scale bar: 20 μm. Inset is a magnified view of the green boxed region of interest. Negative control and histopathology images in Fig S3. **(B)** Nasal swabs were used to perform RT-PCR and the surplus was used for imaging experiments. **(C)** FISH against SARS-CoV-2 RNA in a patient sample tested positive for SARS-CoV-2 by RT-PCR. Scale bar: 20 μm. Negative control in Fig S4. In panels (A) and (C), positive-strand RNA was labeled with Cy3 (white), and nuclei with DAPI (blue).

part of routine care at the Hospital Ambroise Paré (Fig 4B). Samples were screened for the presence of SARS-CoV-2 with RT-PCR. The remainder of the sample not used for diagnostics was deposited on coverglass and we used CoronaFISH to image the positive-strand RNA of SARS-CoV-2. Negative samples (RT-PCR Ct value above detection limit) gave no specific signal, but some areas showed substantially higher background than in the cultured cell lines (Fig S4). This background was rather homogenous, and thus distinct from the RNA signal seen so far in infected cells above. By contrast, in a RT-PCR–positive sample (Ct value = 21), we detected a strong RNA-FISH signal in a subset of cells (Fig 4C). Therefore, although a systematic analysis on many more samples will be required to assess specificity, our data suggests that CoronaFISH probes may allow the detection of SARS-CoV-2 in patient-derived samples in a clinical setting.

### EM visualization of SARS-CoV-2 RNA

Above, we demonstrated how CoronaFISH allows using different fluorescently labeled secondary detector oligos. We reasoned that this flexibility could also enable other imaging modalities including EM. EM is optimally suited to reveal how infection alters the cellular ultrastructure. Indeed, conventional EM images of glutaraldehyde-fixed samples showed a dramatic reorganization of the cytoplasm of Vero cells upon infection with SARS-CoV-2 (Wuhan/IDF strain), characterized by a loss of Golgi stacks (Fig S5A and B) and prominent new features, including numerous DMVs (Fig S5C), which have recently been identified as the main ROs of SARS-CoV-2 (Cortese et al, 2020; Klein et al, 2020; Snijder et al, 2020; Wolff et al, 2020). Budding of viral particles appeared restricted to the lumen of the ER (data not shown) and to electron-lucent vesicles derived from the ER–Golgi intermediate compartment (ERGIC) that were distinct from DMVs (Fig S5D), also in agreement with prior studies (Sicari et al, 2020).

We reasoned that coupling EM with RNA in situ hybridization (EM-ISH) would allow for ultrastructural studies of SARS-CoV-2–infected cells with direct visualization of the vRNA. We previously used EM-ISH to detect various cellular RNAs using DNA- or ribo-probes (Hubstenberger et al, 2017; Yamazaki et al, 2018). Here, we adapted this labeling approach by using the same 96 primary oligos against the positive strand of SARS-CoV-2 as before, but hybridized to a secondary oligo with biotin at its 5′ end. These hybrids were detected with an anti-biotin antibody conjugated to 10-nm gold particles (Fig 5A). EM imaging was performed on thin sections (80 nm) of Lowicryl K4M-embedded Vero cells, either uninfected or infected (MOI 0.1, 36 h p.i.).

In uninfected samples, very few gold particles were detected (Fig 5B). Manual counting on random fields yielded a mean of only 0.5 (± 0.3 s.d.) gold particles/$\mu m^2$ in nuclear and cytoplasmic areas (n = 9 regions, for a total of 94.4 $\mu m^2$), indicative of low background labeling. Although a much larger number of small particles were visible, their size was consistent with ribosomes rather than gold particles (Fig 5E and F). In infected cells, gold particles were visible in large numbers at several locations, notably at intracytoplasmic (Fig 5C) and extracellular viral particles (Fig S5E), as expected. However, DMVs were the most heavily labeled structures (Fig 5D), with manual counts of 180 (± 39 s.d.) gold particles/$\mu m^2$ in DMV

zones, a 360-fold increase over uninfected cells (n = 11 regions, for a total of 11.9 $\mu m^2$). Strikingly, we observed gold particles accumulating along DMV internal 10 nm thick fibers and at the periphery of DMVs (Fig 5D). Although the nature of these fibers remains to be determined, this accumulation of gold particles could reflect a slow export of the viral genomes through the recently described pores spanning the DMV double membrane (Wolff et al, 2020). Finally, gold particles also labeled large lysosomal organelles, shown to play a role in exocytosis of mouse $\beta$-Coronaviruses (Ghosh et al, 2020) and containing densely packed SARS-CoV-2 virions (Fig S5F).

These data demonstrate the flexibility of our probe sets, permitting their use for both fluorescence and EM imaging, and their potential for ultrastructural studies of SARS-CoV-2 replication.

## Discussion

Here, we presented CoronaFISH, an approach based on smiFISH (Tsanov et al, 2016) permitting visualization of the positive and negative RNA strands of SARS-CoV-2. We validated sensitive and specific detection of SARS-CoV-2 RNA by fluorescence microscopy in Vero cells, at different MOIs, and under pharmacological inhibition, in several human cell lines (Caco-2, Huh7, and Calu-3), human lung tissue and nasal swabs. Last, we demonstrated the flexibility of our technique by adapting it for EM imaging of the vRNA.

Our two probe sets each consist of 96 probes, each of which is conjugated to two fluorescent dyes, theoretically enabling 192 fluorescent dyes to target each RNA strand. This results in a very bright signal, allowing vRNA detection in challenging samples, as demonstrated for low MOIs, high Remdesivir concentration and with the tissue and patient samples, where the increased signal intensity helps to distinguish true signal despite high autofluorescence.

Several different variations of FISH/smFISH have been developed over the last years (Pichon et al, 2018), and some of them have also been used to visualize SARS-CoV-2 RNA. Commercial solutions include Stellaris RNA-FISH (LGC Biosearch Technologies) (Lee et al, 2021 Preprint), HuluFISH (PixelBiotech) (Stahl et al, 2021), RNAscope (ACDBio) (Liu et al, 2021), and hybridization chain reaction smFISH (HCR; Molecular Instruments) (Acheampong et al, 2021 Preprint). These methods target specific SARS-CoV-2 RNA regions, such as the S or N gene, or ORF1a. For CoronaFISH, by contrast, we designed 96 probes spaced along the entire ~30 Kb viral genome. These probes are provided individually in a 96-well plate format, therefore subsets of probes against specific regions (e.g., to target-specific viral genes) can be selected individually, instead of using all probes as a complete pool as demonstrated here. The biggest difference between these approaches is in the actual labeling strategy. RNAscope and HCR both provide signal amplification methods, whereas HuluFISH and Stellaris use directly labeled probes. In CoronaFISH, labeling is achieved with secondary detector probes, which can be easily swapped, thereby allowing the use of different fluorophores and simultaneous imaging of positive and negative strands or even changing the imaging modality, as demonstrated by our EM-ISH experiments. This labeling flexibility, together with the fact that our probes target the entire genome, makes CoronaFISH a useful alternative to other FISH-based methods.

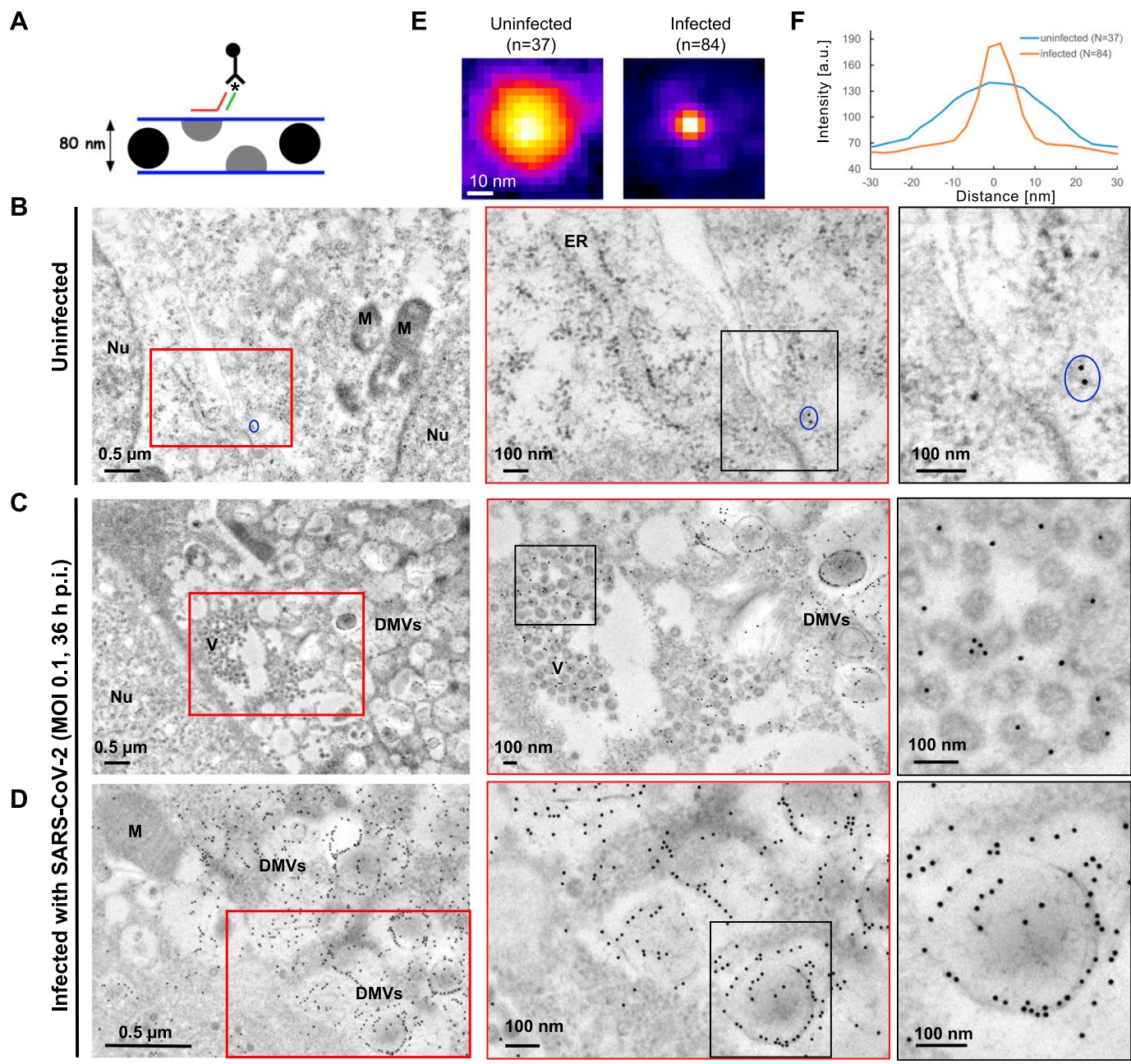

**Figure 5. CoronaFISH for EM.**
**(A)** Principle of EM-ISH performed on thin sections of Lowicryl K4M-embedded infected and uninfected Vero cells. The asterisk indicates biotin at the 5′-end of the secondary oligo, which is recognized by the anti-biotin antibody conjugated to 10-nm gold particles. As sketched, only virions with a section on the upper face of the ~80-nm ultrathin section will be labeled. **(B)** Uninfected control samples. Blue oval surrounds an example of sparse background staining by electron-dense gold particles. The less defined punctate structures such as those lining up the ER (middle panel) are ribosomes (see panels E, F). Nu, nucleus; M, mitochondria; ER, endoplasmic reticulum. **(C, D)** Overviews of SARS-CoV-2-infected Vero E6 cells showing major cytoplasmic vacuolization by virus-induced double-membrane vesicles (DMVs). Gold particles labeling positive strand SARS-CoV-2 RNA can be observed on intracytoplasmic aggregates of viral particles (C). See Fig S5E for extracellular aggregates. DMVs were the most heavily labeled structures, and displayed accumulations of viral RNAs, especially on peripheral 10 nm fibers (D, rightmost frame and Fig S5F). By contrast, mitochondria or nuclei were not significantly labeled. DMV: double-membrane vesicles. Nu, nucleus; M, mitochondria. **(E)** Averaged image from punctate structures in panels (B, C, D) detected with FISH-quant and aligned to the same center (Mueller et al, 2013). **(F)** Line profiles through the averaged spots in (E). The punctate structures visible in infected cells have a size in agreement with the 10 nm diameter of gold particles, whereas the punctate structures in uninfected cells are substantially larger and consistent with 30 nm ribosomes.

Compared with other methods to detect the virus, our hybridization based technique offers several advantages. First, CoronaFISH directly visualizes the viral genome (and/or its replication intermediate) in infected cells rather than viral proteins. This provides a more specific indication of viral presence and replication because structural viral proteins may be found in cells or subcellular compartments where the viral genome is absent or where it does not replicate. Therefore, CoronaFISH could be

instrumental in distinguishing productive from non-productive (abortive) infection, as has been reported, for example, in the context of macrophage infection and antibody-dependent enhancement of SARS-CoV-2 infection (Hui et al, 2020; Grant et al, 2021). Thus, CoronaFISH offers a powerful tool to help define the molecular mechanisms of SARS-CoV-2 pathogenesis, in particular to identify the role of active viral replication in the evolution towards severe disease, infection of immune cells and inflammatory response (Tay et al, 2020). In addition, the ability to distinguish and simultaneously visualize positive and negative RNA strands permits the study of replication kinetics in single cells and to uncover spatiotemporal aspects of the infection cycle. Second, the high specificity of these probes owing to their unique complementarity to the SARS-CoV-2 sequence allows distinguishing it from other related viruses, which can be a problem for antibodies against similar epitopes of different related viral strains. Third, probes can be synthesized within a few days, allowing quick turnover compared with antibody production. Fourth, the CoronaFISH approach is inexpensive. Primary probes can be ordered at low cost, and the provided quantities (smallest synthesis scale provides nanomoles for each oligo) suffice for thousands of experiments. This makes CoronaFISH attractive for high throughput image-based screening of large libraries of antiviral compounds, as illustrated by our Remdesivir experiment.

CoronaFISH can also be used in combination with immunofluorescence for the detection of viral or host proteins (Rensen et al, 2021) and is compatible with GFP stains (Tsanov et al, 2016). RNA-FISH and IF combined have also been shown to be suitable for flow cytometry and FACS (Arrigucci et al, 2017). More complex implementations enable multiplexed detection of multiple RNA species (Pichon et al, 2018) and will thus permit probing the host–pathogen interaction at the single-cell-single-virus level.

Compared with single-cell RNA-seq, CoronaFISH provides information on single cells in their spatial context because experiments do not require cell dissociation. Our approach can thus deliver insights into the viral life-cycle, including occurrence and abundance of positive and negative-strand RNA, their subcellular localization, and their interplay with the host and with structures induced by SARS-CoV-2 infection (DMVs, replication compartments, and ERGIC). CoronaFISH also provides a unique tool to study virus–host interactions in tissue. Furthermore, studying vRNA presence in thousands of cells is possible by using automated image analysis, and can hence enable the detection of rare events. This will allow examining the effects of infection on cellular aspects such as cell morphology, cell fusion, cell-to-cell transmission, or tissue (re)organization.

Our data on nasal swabs suggest that CoronaFISH may be used on clinical samples and potentially as the basis for diagnostic tests. Unlike standard RT-PCR tests, CoronaFISH does not require RNA extraction or enzymatic reagents, which have at times been in short supply. Because our probes span the whole length of the ~30 Kb SARS-CoV-2 genome, they should enjoy higher robustness against mutations or partial RNA degradation than methods that target one or a few genes. This may be useful for diagnostics to circumvent the limitations of PCR-based tests because, for example, the alpha variant of SARS-CoV-2 has been reported to yield negative results in some PCR tests based on the Spike gene (https://www.fda.gov/medical-devices/coronavirus-covid-19-and-medical-devices/sars-cov-2-viral-mutations-impact-covid-19-tests). Indeed, application of CoronaFISH to Vero cells

infected with alpha or beta variants confirms robust visualization of these variants despite their sequence differences (Fig S6).

The cost of CoronaFISH reagents also compares favorably to those used in standard PCR assays. Despite these advantages, a diagnostic test based on CoronaFISH would face two hurdles: slowness and the need for a fluorescence microscope. The duration of the FISH experiment (~2 d) is currently too long for a rapid test. However, microfluidic devices can be used to reduce this delay to less than 15 min, comparable to fast antigenic tests (Shaffer et al, 2015). The requirement for a fluorescence microscope may also be alleviated using cheap do-it-yourself imaging systems, for example, smartphones combined with inkjet-printed lenses (Cybulski et al, 2014; Sung et al, 2017). Combined with CoronaFISH, such portable and low-cost imaging systems could potentially facilitate point-of-care diagnostics.

To conclude, we believe that the probes and complementary labeling approaches described here expand the toolbox for studying SARS-CoV-2 and hope that the resources provided (sequences, protocol, and source code) will facilitate their adoption by the community to better understand, diagnose and fight this virus.

# Materials and Methods

### Probe design

The entire code for probe design is available on GitHub at: https://github.com/muellerflorian/corona-fish. The probe design involves several steps to ensure high sensitivity for the detection of SARS-CoV-2 RNA, while minimizing false-positive detection of other $\beta$-coronaviruses, other pathogens provoking similar symptoms, and transcripts of the host organism. Below, we list in parentheses how many probes remain after each selection step for probes targeting positive and negative strands, respectively.

The initial list of probes for the positive and negative strands was generated with Oligostan (N = 615 and 608, respectively) (Tsanov et al, 2016). We then selected all probes with a GC content between 40% and 60% and probes satisfying at least two out of five previously established criteria for efficient oligo design (Xu et al, 2009) (N = 385/362).

To guarantee that the probes are insensitive towards known mutations of SARS-CoV-2 (as of March 2020), we selected only probes with not more than two mismatches with any of 2,500 aligned SARS-CoV-2 sequences (N = 333/311).

We then performed a local BLAST against other $\beta$-coronaviruses (SARS, MERS, HKU1, OC43, NL63, or 229E), other pathogens and viruses causing similar symptoms (*Mycobacterium tuberculosis*, human parainfluenza virus type 1–4, respiratory syncytial virus, human metapneumovirus, *Mycoplasma pneumoniae*, *Chlamydophila pneumoniae*, influenza A–D, rhinovirus/enterovirus), as well as the transcriptome of the most common host organisms (*Homo sapiens*, *Mus musculus*, African Green monkey, hamsters, and ferret). We excluded all probes with more than 22 matches in any of these BLAST searches (N = 115/114).

Last, we selected the 96 probes with the highest NGS coverage. Probe sequences are provided in Table S1.

## CoronaFISH

To visualize vRNA molecules, we used the smiFISH approach (Tsanov et al, 2016). Unlabelled primary probes are designed to target the RNA of interest and can be prehybridized with fluorescently labeled secondary detector oligonucleotides for visualization. Probes were designed as described above.

A detailed protocol is available in Supplemental Data 1. Cells were fixed in 4% PFA for 30 min, washed twice with PBS++, and stored in nuclease-free 70% ethanol at −20°C until labeling. On the day of the labeling, the samples were brought to room temperature, washed twice with wash buffer A (2× SSC in nuclease-free water) for 5 min, followed by two washing steps with washing buffer B (2X SSC and 10% formamide in nuclease-free water) for 5 min. The target-specific primary probes were prehybridized with the fluorescently labeled secondary probes via a complementary binding readout sequence. The reaction mixture (10 $\mu$L) contained 40 pmol of primary probes, and 50 pmol of secondary probes in 1X NEBuffer buffer (New England Biolabs). Prehybridization was performed in a PCR machine with the following cycles: 85°C for 3 min, followed by heating to 65°C for 3 min, and a further 5 min heating at 25°C. 2 $\mu$l of this FISH probe stock solution was added to 100 $\mu$l of hybridization buffer (10% [wt/vol] dextran, 10% formamide, and 2X SSC in nuclease-free water).

Samples were placed on parafilm in a humidified chamber on 100 $\mu$l of hybridization solution, sealed with parafilm, and incubated overnight at 37°C. The next day, cells were washed in the dark at 37°C without shaking for >30 min twice with prewarmed washing buffer B. Sample were washed once with PBS for 5 min, stained with DAPI in PBS (1:10,000) for 5 min, and washed again in PBS for 5 min. Samples were mounted in ProLong Gold antifade mounting medium.

## CoronaFISH combined with immunofluorescence

Cells were prepared as described above. Before hybridization, cells were washed with buffer B supplemented with 10% normal goat serum and RNASecure 1/1,000 (AM7005; Invitrogen) for 10 min at room temperature. CoronaFISH probes were prehybridized as described above. 2 $\mu$l of this prehybridized probe solution and J2 antibody (Scicons.eu) (1/200) was added to 100 $\mu$l of hybridization buffer (10% (wt/vol) dextran, 10% formamide, 2X SSC, 0.5% BSA, and 1/1,000 RNAsecure in nuclease-free water). Samples were placed on parafilm in a humidified chamber on 100 $\mu$l of hybridization solution, sealed with parafilm, and incubated for 4 h at 37°C. Cells were washed in the dark at 37°C without shaking for >30 min twice with pre-warmed washing buffer B, 1/1,000 RNAsecure. Secondary antibody (Goat anti-mouse labeled with Cy3 [1/500]) was added to hybridization buffer (100 $\mu$l of 2XSSC, 0.5% BSA, 1/1,000 RNASecure). Samples were placed on parafilm placed in a humidified chamber on 100 $\mu$l of hybridization buffer, and incubated 30 min at room temperature. Samples were washed twice with 2X SSC in the dark at 37°C with shaking for >10 min. Samples were washed once with 2X SSC with DAPI (1: 1,000) for 10 min, and washed again in 2X SSC for 5 min. Samples were mounted in ProLong Gold antifade mounting medium.

## Infection of cell lines

### Virus strains

The viral stocks used originate from hCoV-19/France/IDF0372/2020 (Wuhan/IDF strain) and hCoV-19/France/GE1973/2020 (D614G). The human sample from which hCoV-19/France/GES-1973/2020 was isolated was provided by Pr. Laurent Andreoletti, CHU of Reims. For the experiment using SARS-CoV-2 variants (Fig S6), the following strains were used: alpha (B.1.1.7) hCoV-19/France/IDF-IPP11324/2021 and beta (B.1.351) hCoV-19/France/IDF-IPP00078/2021. The human sample from which strain hCoV-19/France/IDF-IPP11324/2021 was isolated has been provided by Dr. Pierre-Yves Leonard, Laboratoire Laborizon Maine Anjou, and the human sample from which hcoV-19/France/IDF-IPP00078/2021 was isolated has been provided by Dr. Mounira Smati-Lafarge, CHI of Créteil. All the strains were kindly gifted by the National Reference Center for Respiratory Viruses hosted by Institut Pasteur and headed by Prof. Sylvie van der Werf.

### Vero cells

We used both Vero and Vero E6 cells for our experiments. Vero cells were used for experiments with the SARS-CoV-2 Wuhan/IDF strain. On the day before the infection, Vero E6 cells were trypsinized and diluted in DMEM–Glutamax 10% FBS. They were then seeded, 8 × $10^4$/well, in a 12-multiwell with coverslips. The day of the infection the medium of the cells was discarded and the monolayers were infected with SARS-CoV-2 (Wuhan/IDF strain) virus in DMEM–GlutaMAX 2% FBS for 1 h at 37°C 5% $CO_2$ at a MOI of 0.02. After the desired infection duration, the supernatant was collected for virus titration, and the cells were washed with PBS++, fixed with 4% EM-grade PFA for 30 min at RT, and processed for smiFISH.

For the experiment using different MOIs, Vero E6 cells were infected for 2 h at 37°C 5% $CO_2$ in DMEM/1% FBS with SARS-CoV-2 D614G strain at MOI of 10-1-0.1. After infection, the inoculum was removed and the cells were incubated in DMEM/5% FBS for 24 h. For the experiment with alpha and beta variants, Vero E6 cells were infected at MOI of 0.1 and were fixed 29 h p.i.

### Vero cells for EM-ISH

On the day before the infection, Vero cells were trypsinized and diluted in DMEM–GlutaMAX 10% FBS. They were then seeded, 7 × $10^5$ cells/well, in a six multiwell plate. On the day of the infection, the medium of the cells was discarded and cells were infected with SARS-CoV-2 virus in DMEM–GlutaMAX 2% FBS at MOI of 0.1. After 36 h, the supernatant was collected for virus titration and the cells were washed with PBS++. The monolayers were fixed using 4% EM-grade PFA in 0.1M Sorensen's buffer for 1 h at 4°C.

### Human cell lines

On the day before the infection, Huh7, Caco-2, Calu-3, and Vero E6 cells were trypsinized and diluted in DMEM–GlutaMAX 10% FBS (Huh7 and Vero E6), MEM 20% FBS + NEAA, sodium pyruvate and GlutaMAX (Caco-2), and RPMI 20% FBS + NEAA (Calu-3). After 6 h, the medium of the cells was discarded and the monolayers were infected with SARS-CoV-2 virus (Wuhan/IDF strain) in triplicate at an MOI of 0.2 in DMEM–GlutaMAX 2% FBS. After 36 h, the supernatant was collected for virus titration and the cells were washed with

PBS++. The cells were fixed using 4% EM-grade PFA for 30 min at RT and processed for smiFISH.

### Titration protocol (focus forming assay)

Vero E6 cells were seeded in 96-multi wells at $2 \times 10^4$ cells/well in DMEM–GlutaMAX 10% FBS. The following day the supernatants to be titered were thawed and serially diluted in 10-fold steps in DMEM–GlutaMAX 1% FBS. 100 $\mu$l of the dilutions were used to infect Vero cells growing as a monolayer, and incubated for 2 h at 37°C and 5% $CO_2$. The infection medium was then discarded and a semi-solid media containing MEM 1X, 1.5% CMC, and 10% FBS was added to the monolayers. The cells were incubated at 37°C and 5% $CO_2$ for 36 h. Cells were then fixed with 4% formaldehyde and foci were revealed using a rabbit anti-SARS-CoV N antibody (for experiments with the SARS-CoV-2 Wuhan/IDF strain) or a rabbit anti-SARS-CoV-2 N antibody (both gifts from N Escriou from Institut Pasteur) and matching secondary HRP-conjugated secondary antibodies. Foci were visualized by DAB staining and counted using an Immunospot S6 Analyser (Cellular Technology Limited CTL). Viral titers were expressed as focus forming units (FFU)/ml.

### Inhibitors assay

To determine IC50 of Remdesivir in our system, Vero E6 cells were pre-treated with serial dilutions of Remdesivir (100 nM - 100 $\mu$M) for 1 h at 37°C in DMEM/1% FBS. The cells were then infected with SARS-CoV-2 D614G at MOI 0.1 for 2 h, and after removal of the inoculum maintained in DMEM/5% FBS containing the different concentrations of Remdesivir for 2 d. Supernatant was then collected and titered by focus forming assay. IC50 values were calculated by nonlinear regression analysis (log(inhibitor) versus response–Variable slope (four parameters)) using Prism 6, GraphPad software. For the FISH experiment, Vero E6 cells infected with SARS-CoV-2 D614G virus at MOI 0.001 were treated with Remdesivir dose range 0.86–10 $\mu$M and fixed 24 h p.i.

## Image analysis of infected cell lines

Nuclei were automatically segmented in 2D images with an ImJoy (Ouyang et al, 2019) plugin using the CellPose model (Stringer et al, 2021). Source code for segmentation is available here: https://github.com/fish-quant/segmentation. Equidistant regions with a width of 50 pixels were calculated around each nucleus. Overlapping regions from different nuclei were removed. 3D FISH images were transformed into 2D images with a maximum intensity projection along z. Signal intensity for each cell was determined as the 99% quantile of all pixels in the equidistant region around its nucleus.

## Lung tissue

Lung autopsy material from the COVID-19 patient was provided by the human biological sample bank of the Lille COVID working group "LICORN." The use of this autopsy material for research purposes was approved by local ethics review committees at Lille Hospital. Lung autopsy material from the control patient with diffuse alveolar damage prior to the COVID-19 pandemic was provided by the Pathological department of Hôpital Necker Enfants Malades.

Lung autopsy material was fixed in 10% neutral buffered formalin and paraffin embedded, 4-$\mu$m sections were stained with hematoxylin and eosin for histological analysis using light microscopy. The tissue was then de-paraffinized and SARS-CoV-2 RNA was detected by FISH as described above.

## Nasal swab patient samples

Respiratory specimens (nasal swabs) have been collected from patients with respiratory symptoms as part of routine care at the Hospital Ambroise Paré in late March 2020. No additional samples were collected in the course of this work. Patients were contacted, informed about the research project, and given the possibility to oppose the use of their samples for this project. Lack of opposition to participate in clinical research was verified in the records of all patients whose samples were used here. This project has been recorded in the French public register Health-data-Hub (no. F20200717122429). Processing of personal data for this study complies with the requirements of the "reference methodology MR-004" established by the French Data Protection Authority (CNIL) regarding data processing in health research.

Samples were screened for the presence of SARS-CoV-2 with RT-PCR or processed for CoronaFISH as described below.

### RNA extraction

400 $\mu$l of clinical samples were extracted in 300 $\mu$l of elution buffer (Total NA Lysis/Binding buffer) for 20 min at room temperature with gentle agitation. RNA was extracted with the MagNA Pure compact (Roche) and the MagNA Pure Compact DNA Isolation Kit I (Roche) following the protocol "Total_NA_Plasma_external_lysis purification protocol." Final dilution of RNA was in 50 $\mu$l elution buffer.

### RT-PCR

Screening for SARS-CoV-2 was performed by RT-PCR following a modified protocol recommended by the French National Reference Center for Respiratory Viruses, Institut Pasteur, Paris using Ag Path-ID One-Step RT-PCR kit (Thermo Fisher Scientific). PCR reaction was run on the ABI PRISM 7900 system (Applied Biosystems) with the following cycle settings: 45°C 10'; 95°C 5'; 45 X (95°C 15''; 58°C 45''). Primer sequences and concentration are provided in Table S2.

### CoronaFISH

Thin-layered samples on a cover-slide suitable for FISH were obtained with a Cytospin protocol. 150 $\mu$l of the sample were deposited in a Cytofunnel (1102548; Thermo Fisher Scientific). Samples were then centrifuged at 72$g$ for 10 min at room temperature with a Cytospin 4 cytocentrifuge (Thermo Fisher Scientific) on Cytoslides (5991059; Thermo Fisher Scientific). Cells were fixed in PBS-PFA 4% for 30 min and then conserved frozen in 70% ethanol at −20°C. CoronaFISH labeling was performed with Cy3-labeled plus-strand probes as described above, with one exception: for hybridization

400 µl hybridization buffer was used per sample instead of 100 µl (with the same final probe concentration).

### EM-ISH

#### *Fixation and embedding for EM*

For Epon embedding, cell cultures were fixed for 1 h at 4°C in 2% glutaraldehyde (Taab Laboratory Equipment) in 0.1 M phosphate buffer, pH 7.3. During fixation, cells were scraped off from the plastic substratum and centrifuged at 5,000*g* for 15 min. Cell pellets were dehydrated in increasing concentrations of ethanol and embedded in Epon. Polymerization was carried out for 48 h at 64°C. Ultrathin sections were collected on Formvar–carbon–coated copper grids (200 mesh) and stained briefly with standard uranyl acetate and lead citrate solutions.

Embedding in Lowicryl K4M (Chemische Werke Lowi) was carried out on Vero cells fixed either in 4% formaldehyde (Merck) or in 2% glutaraldehyde at 4°C. Cell pellets were equilibrated in 30% methanol and deposited in a Leica EM AFS2/FSP automatic reagent handling apparatus (Leica Microsystems). Lowicryl polymerization under UV was for 40 h at –20°C followed by 40 h at +20°C. Ultrathin sections of Lowicryl-embedded material were collected on Formvar–carbon–coated gold grids (200 mesh) and stored until use.

#### *EM-ISH*

For EM-ISH, the SARS-CoV-2 RNA (positive strand) probe was composed of the same set of 96 oligodeoxynucleotides that was used for RNA-FISH. The secondary oligonucleotide, however, was modified by a custom-added biotin residue at its 5′-end (QIAGEN). Hybrids of the SARS-CoV-2 RNAs with the probe were detected with a goat anti-biotin antibody conjugated to 10 nm gold particles (BBI International).

High-resolution in situ hybridization was carried out essentially as described previously (Hubstenberger et al, 2017; Yamazaki et al, 2018). The hybridization solution contained 50% deionized formamide, 10% dextran sulfate, 2 × SSC, and a final concentration of 80 ng/ml of a mix of 1 µg/µl primary oligonucleotides and 1.2 µg/µl of biotinylated secondary oligonucleotide stored at –20°C. EM grids, with ultrathin sections of either formaldehyde- or glutaraldehyde-fixed cells side down, were floated for 3 h at 37–42°C on a 1.5-µl drop of hybridization solution deposited on a parafilm in a moist glass chamber. EM grids were then briefly rinsed over three drops of PBS and incubated 30 min at RT on a drop of goat anti-biotin antibody (BBI International) conjugated to 10 nm gold particles diluted 1/25 in PBS. EM-grids were further rinsed on two drops of PBS and finally washed with a brief jet of deionized water at high intensity. After a 10-min drying on filter paper with thin sections on top, the EM grids were stained for 1 min on a drop of 4% uranyl acetate in water and dried on filter paper 30 min before observation under the EM. Standard lead citrate staining was omitted to favor higher contrast of gold particles over the moderately stained cellular structures.

Sections were analyzed with a Tecnai Spirit transmission electron microscope (FEI), and digital images were taken with an SIS MegaviewIII charge-coupled device camera (Olympus). Quantitation was performed by manually counting gold particles on surfaces that were measured using analysis software (Olympus Soft Imaging Solutions). Statistics were calculated using Excel (Microsoft).

## Supplementary Information

## Acknowledgements

We would like to thank Edouard Bertrand, who originally developed smiFISH and Hans Johansson for insightful discussions. We also thank Guillaume Dumenil (Ultrastructural Bioimaging UTechS of Institut Pasteur) for having suggested the application of CoronaFISH to EM and for follow-up discussions. We thank Nathalie Jolly and Nathalie Clément (Center for Translational Science, Institut Pasteur) and Cloé Giquel (Legal department, Institut Pasteur) for their help in obtaining authorization to use patient samples. E Simon-Loriere acknowledges funding from the French Government's Investissement d'Avenir program, "INCEPTION' (ANR-16-CONV-0005). G Barba-Spaeth acknowledges funding by the Institut Pasteur Coronavirus task force (don AXA COVID-19 project COVID-SPREAD). C Zimmer acknowledges funding by Fondation pour la Recherche Médicale (Equipe FRM, DEQ 20150331762), Institut Carnot Pasteur MS, and Institut Pasteur.

### Author Contributions

E Rensen: conceptualization, validation, investigation, and methodology.
S Pietropaoli: investigation.
F Mueller: conceptualization, software, formal analysis, methodology, supervision, and writing—original draft, review, and editing.
C Weber: validation, investigation, and visualization.
S Souquere: investigation and visualization.
S Sommer: resources, investigation, and visualization.
P Isnard: resources.
M Rabant: resources.
J-B Gibier: resources.
F Terzi: resources and methodology.
E Simon-Loriere: resources, software, and data curation.
M-A Rameix-Welti: resources, methodology, review, and editing.
G Pierron: investigation, supervision, and methodology.
G Barba-Spaeth: resources, supervision, validation, investigation, review, and editing.
C Zimmer: conceptualization, resources, supervision, and writing—original draft, review, and editing.

### Conflict of Interest Statement

The authors declare that they have no conflict of interest.

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
