## [Reviewer comments · Life Science Alliance]

Life Science Alliance

Sensitive visualization of SARS-CoV-2 RNA with CoronaFISH

Elena Rensen, Stefano Pietropaoli, Florian Mueller, Christian Weber, Sylvie Souquere, Sina Sommer, Pierre Isnard, Marion Rabant, Jean-Baptiste Gibier, Fabiola Terzi, Etienne Simon-Lorière, Marie-Anne Rameix-Welti, Gerard Pierron, Giovanna Barba-Spaeth, and Christophe Zimmer

DOI: <https://doi.org/10.26508/lsa.202101124>

Corresponding author(s): *Christophe Zimmer, Institut Pasteur and Florian Mueller, Institut Pasteur*

Review Timeline:

Submission Date:	2021-05-27
Editorial Decision:	2021-05-27
Revision Received:	2021-11-10
Editorial Decision:	2021-11-15
Revision Received:	2021-12-22
Accepted:	2021-12-22

Scientific Editor: Novella Guidi

Transaction Report:

Please note that the manuscript was previously reviewed at another journal and the reports were taken into account in the decision-making process at *Life Science Alliance*.

Referee #1 Review

Report for Author:

The manuscript by Ransen E. from the group of Christoph Zimmer describes a new tool to directly visualize SARS-CoV-2 RNA in infected cells by fluorescence in situ hybridization (FISH) using probes for positive or negative RNA strand. The authors also demonstrate the adaptation of these probes to electron microscopy (EM).

FISH represents a valid method for specific visualization of viral RNA, and the authors here provide a pipeline for design of probes and a methodology for visualization in cell lines, as well as in tissues from infected patients.

I think that authors should add some additional controls:

In the setting of the method the authors should use several different MOIs to validate the detection rate by FISH. They should also confirm the amounts of viral RNA by PCR.

Also, they should use the dsRNA antibody (J2) to verify the presence of viral RNA. This control should be especially useful on lung and other tissues, where there seems to be a high accumulation of FISH probes at certain sites.

In the discussion part they say that because they choose probes to cover the entire genome, their method is likely to be more robust to mutations- that is a disputable conclusion as there is no evidence that FISH can be more precise than the PCR. They should either try to show it or to reformulate their statement.

Referee #2 Review

Report for Author:

Ransen et al. report the visualization of SARS-CoV-2 RNA in infected cell culture, nasal swabs, and tissues by using a sophisticated Fluorescence In-Situ Hybridization technique (FISH). Analysis of SARS-CoV-2 replication is an important research field, and FISH allows a direct and sensitive visualization of viral RNAs e.g. for patient samples or sample materials from animal

trials. The newly developed and validated FISH-protocol (CoronaFISH) allows the sensitive detection and differentiation of both positive and negative sense SARS-CoV-2 RNA. The technique was established and tested with different materials, including infected cell cultures and tissues. Finally, CoronaFISH was also established as a protocol for electron microscopy.

The paper is strongly methodological and focuses almost entirely on method development, establishment, and a first validation with selected sample materials. The method description is on a very high level, and the functionality is impressively demonstrated with corresponding figures e.g. of the different stainings including electron microscopy. Most novelty is provided by the transfer of the technique to electron microscopy.

The paper is overall well written and provides all necessary detailed information to transfer the described detection system.

Nevertheless, there are also major concerns with the manuscript:

1. There are several studies available since beginning of 2020 describing the use of in situ hybridization for detection of SARS-CoV-2-RNA. Some of these also describe the differentiation of positive and negative strand RNA (e.g. Liu et al. 2020, JCI Insight. doi: 10.1172/jci.insight.139042.). Numerous other studies used FISH methods including e.g. RNAscope technology for detection of SARS-CoV-2 in tissue samples. Furthermore, there is now commercially available probe sets for sophisticated implementation of SARS-CoV-2 FISH analyses (e.g. from BioCat or Metasystems). Some of the available protocols use also ORF1-specific probes in addition to the spike encoding region.

2. The methodological focus of the study is very dominant. The hypothesis-oriented use of the established CoronaFISH technique is not clearly visible. Therefore, it is not fully clear what particular benefit the established protocols have compared to the other available hybridization methods.

Minor concerns:

Introduction section:

- Line 40: an update of the case numbers is necessary and the data of data collection should be provided
- Line 81: the term smiFISH has to be explained here
- Other available in situ techniques should be mentioned and also the availability of commercial systems.

Results section:

- Line 134: is the viral load extremely high or only the level of viral RNAs? This should be clarified.
- What are the SARS-CoV-2-genome copy numbers per 1000 cells?
- 143: What is the reason for the statement „...also reflecting diminished labelling efficacy...“

Discussion section:

- Line 313: the advantage of detection of RNA instead of proteins has to be explained in more detail.
- What is the advantage in relation to other available hybridization systems?
- Line 318/319: what kind of molecular mechanisms of SARS-CoV-2 pathogenesis can be defined? More information and data are missing here. What analyses are improved by CoronaFISH in comparison to other in situ technologies?
- What is the difference to techniques like RNAscope which are used in numerous studies for SARS-CoV-2 detection.

Overall, the study seems particularly suitable for a method-oriented journal.

Referee #3 Review

Report for Author:

The paper titled "Sensitive visualization of SARS-CoV-2 RNA with CoronaFISH" describes the utilization of FISH to detect viral RNA of SARS-CoV-2 from cell lines, human tissue and patient samples. The author also adapted the probes to electron microscopy. The FISH method provides a complement way of Covid-19 detection though it is time-consuming and needs more complicated machines (e.g. microscope). This paper is well written, clear and technically sound. However, overall, CoronaFISH is not conception novel and it raises couples of concerns.

Major points:

- 1, Remdesivir is used to inhibit the RdRp activity and does not interfere with the entry of SARS-CoV-2. Can CoronaFISH be exploited to detect vRNA in cytosol before its replication?
- 2, N protein usually binds viral RNA genome and I am wondering whether N protein can dramatically decrease the sensitivity of CoronaFISH, especially when it was used to detect the viral particles (for example, samples from environment).
- 3, The authors would better compare the sensitivity of CoronaFISH with IF (anti N protein or S protein) to support their claim.
4. The detection limit of CoronaFISH for the samples from nasal swabs should be provided and compared with other methods such as QPCR and CRISPR.
5. In the inhibition assay, the reproducibility of CoronaFISH should be provided. And the IC50 detected by CoronaFISH could be

compared with other methods.

6. More data in detecting replication kinetics by CoronaFISH should be provided to support the author's statement. Have the authors tried to detect the entry and replication process of SARS-Cov-2 RNA by CoronaFISH with a time span?

Minor points :

1 , line 241, "samples will be required to assess specificity and specificity" should be " samples will be required to assess specificity".

2, Figure 4, b-f should be shown in sequential order in the paper.

3. Did all 96 probes be verified individually?

May 27, 2021

Re: Life Science Alliance manuscript #LSA-2021-01124-T

Dr. Christophe Zimmer
Institut Pasteur
Unité Imagerie et Modélisation
25, rue du Docteur Roux
Paris 75015
France

Dear Dr. Zimmer,

Thank you for submitting your manuscript entitled "Sensitive visualization of SARS-CoV-2 RNA with CoronaFISH" to Life Science Alliance.

For a brief overview, this manuscript was previously reviewed at an LSA partner journal. The authors shared the manuscript and accompanying reviews with LSA editor, Dr. Shachi Bhatt, who found these data interesting and is willing to consider a revised version of this manuscript addressing the following:

- + additional controls requested by Rev 1
- + data about the detection limit and reproducibility of CORONAFish are included (Rev3 pt 4, 5)
- + text revision to expand on the benefit provided by this hybridization method over previously published hybridization methods (Rev 2)

As some new data will be included in the revised manuscript, LSA might reach out to some of the original referees to review the revised manuscript, and walk them through the transfer process, if needed.

To upload the revised version of your manuscript, please log in to your account: <https://lsa.msubmit.net/cgi-bin/main.plex>
You will be guided to complete the submission of your revised manuscript and to fill in all necessary information. Please get in touch in case you do not know or remember your login name.

Thank you for this interesting contribution to Life Science Alliance. We are looking forward to receiving your revised manuscript.

Sincerely,

Shachi Bhatt, Ph.D.
Executive Editor
Life Science Alliance
<http://www.lsajournal.org>
Tweet @SciBhatt @LSAjournal

- A letter addressing the reviewers' comments point by point.
- An editable version of the final text (.DOC or .DOCX) is needed for copyediting (no PDFs).

B. MANUSCRIPT ORGANIZATION AND FORMATTING:

We thank the reviewers for the comments. Our replies are in blue.

Referee #1

The manuscript by Ransen E. from the group of Christoph Zimmer describes a new tool to directly visualize SARS-CoV-2 RNA in infected cells by fluorescence in situ hybridization (FISH) using probes for positive or negative RNA strand. The authors also demonstrate the adaptation of these probes to electron microscopy (EM). FISH represents a valid method for specific visualization of viral RNA, and the authors here provide a pipeline for design of probes and a methodology for visualization in cell lines, as well as in tissues from infected patients.

I think that authors should add some additional controls:

R1.1

In the setting of the method the authors should use several different MOIs to validate the detection rate by FISH. They should also confirm the amounts of viral RNA by PCR.

To address this request, we infected Vero cells at various multiplicities of infection (MOI): 0.001, 0.01 and 0.1. We used focal forming assays to quantify infectious virus in the cell supernatant (instead of PCR, because we are interested in measuring productive infection). We used CoronaFISH to image these cells and found that the proportions of highly infected cells increase with MOI, as expected for specific detection of SARS-CoV-2. These results are presented in the new **Figure 2** (panels **a-e**) and the new **Figure S2** (panels **a-c**).

R 1.2

Also, they should use the dsRNA antibody (J2) to verify the presence of viral RNA. This control should be especially useful on lung and other tissues, where there seems to be a high accumulation of FISH probes at certain sites.

To address this request, we used J2 antibodies against double-stranded RNA in infected Vero cells concomitantly with CoronaFISH and found that CoronaFISH positive cells are also J2-positive and vice-versa, again confirming specific detection. However, we did not repeat this analysis on lung samples, which are in limited supply. These results are shown in **Figure 1f**.

R 1.3

In the discussion part they say that because they choose probes to cover the entire genome, their method is likely to be more robust to mutations - that is a disputable conclusion as there is no evidence that FISH can be more precise than the PCR. They should either try to show it or to reformulate their statement.

Our statement about the higher robustness of CoronaFISH to mutations was simply based on the fact that our probes target the entire 30 Kb viral genome, which makes the labeling less sensitive to mutations than approaches that target a single genomic region, especially regions that are hotspots of mutations like the Spike gene. As we mentioned, mutations have been reported to be a problem for some PCR tests (see <https://www.fda.gov/medical-devices/coronavirus-covid-19-and-medical-devices/sars-cov-2-viral-mutations-impact-covid-19-tests>). To test this experimentally, we applied CoronaFISH to two

variants of SARS-CoV-2 (alpha and beta) and in both cases observed a clear signal, both for the positive and negative RNA strands, thus confirming our claims of robustness to mutations. See our new **Figure S6**.

Referee #2:

Rensen et al. report the visualization of SARS-CoV-2 RNA in infected cell culture, nasal swabs, and tissues by using a sophisticated Fluorescence In-Situ Hybridization technique (FISH). Analysis of SARS-CoV-2 replication is an important research field, and FISH allows a direct and sensitive visualization of viral RNAs e.g. for patient samples or sample materials from animal trials. The newly developed and validated FISH-protocol (CoronaFISH) allows the sensitive detection and differentiation of both positive and negative sense SARS-CoV-2 RNA. The technique was established and tested with different materials, including infected cell cultures and tissues. Finally, CoronaFISH was also established as a protocol for electron microscopy.

The paper is strongly methodological and focuses almost entirely on method development, establishment, and a first validation with selected sample materials. The method description is on a very high level, and the functionality is impressively demonstrated with corresponding figures e.g. of the different stainings including electron microscopy. Most novelty is provided by the transfer of the technique to electron microscopy.

The paper is overall well written and provides all necessary detailed information to transfer the described detection system.

Nevertheless, there are also major concerns with the manuscript:

R 2.1

There are several studies available since beginning of 2020 describing the use of in situ hybridization for detection of SARS-CoV-2-RNA. Some of these also describe the differentiation of positive and negative strand RNA (e.g. Liu et al. 2020, JCI Insight. doi: 10.1172/jci.insight.139042.). Numerous other studies used FISH methods including e.g. RNAscope technology for detection of SARS-CoV-2 in tissue samples. Furthermore, there is now commercially available probe sets for sophisticated implementation of SARS-CoV-2 FISH analyses (e.g. from BioCat or Metasystems). Some of the available protocols use also ORF1-specific probes in addition to the spike encoding region.

The methodological focus of the study is very dominant. The hypothesis-oriented use of the established CoronaFISH technique is not clearly visible. Therefore, it is not fully clear what particular benefit the established protocols have compared to the other available hybridization methods.

In the revised Discussion, we better acknowledge these and other FISH approaches and contrast them more clearly with CoronaFISH.

Minor concerns:

Introduction section:

- Line 40: an update of the case numbers is necessary and the data of data collection should be provided

We updated the case number and added the date.

- Line 81: the term smiFISH has to be explained here

smiFISH stands for "single molecule inexpensive FISH" and was introduced by us in Tsanov et al. (<https://academic.oup.com/nar/article/44/22/e165/2691336>).

- Other available in situ techniques should be mentioned and also the availability of commercial systems.

See response to comment R2.1.

Results section:

- Line 134: is the viral load extremely high or only the level of viral RNAs? This should be clarified.

Our system detects viral RNAs. We have corrected the sentence accordingly.

- What are the SARS-CoV-2-genome copy numbers per 1000 cells?

Estimating genome copy numbers from the images is extremely challenging, as this would require a careful calibration of the intensity of single genomes, and is outside the scope of the present paper.

- 143: What is the reason for the statement „...also reflecting diminished labelling efficacy...“

Although the weaker intensity of the negative strand FISH signal is consistent with previous reports that negative viral RNA is less abundant than positive RNA, we also cannot exclude that the weaker signal of the negative strand may be due to reduced accessibility to the probes. We rephrased the sentence accordingly.

Discussion section:

- Line 313: the advantage of detection of RNA instead of proteins has to be explained in more detail
The main advantage of detecting RNA is that we directly visualize the genome as opposed to one of its products. This is important because it allows to detect ongoing replication of the virus, whereas the visualization of structural viral proteins does not necessarily imply active replication and can reflect unproductive infection. We make this clearer in the revised Discussion.

- What is the advantage in relation to other available hybridization systems?

See response to comment R2.1 above

- Line 318/319: what kind of molecular mechanisms of SARS-CoV-2 pathogenesis can be defined? More information and data are missing here. What analyses are improved by CoronaFISH in comparison to other in situ technologies?

Here we are referring to the ability of CoronaFISH to distinguish productive from unproductive infection. Concerning the comparison with other in situ technologies, see response to comment R2.1 above.

- What is the difference to techniques like RNAscope which are used in numerous studies for SARS-CoV-2 detection.

Again, see our response to comment R2.1 above.

Overall, the study seems particularly suitable for a method-oriented journal.

Referee #3

The paper titled "Sensitive visualization of SARS-CoV-2 RNA with CoronaFISH" describes the utilization of FISH to detect viral RNA of SARS-CoV-2 from cell lines, human tissue and patient samples. The author also adapted the probes to electron microscopy. The FISH method provides a complement way of Covid-19 detection though it is time-consuming and needs more complicated machines (e.g. microscope). This paper is well written, clear and technically sound. However, overall, CoronaFISH is not conception novel and it raises couples of concerns.

Major points:

R 3.1

Remdesivir is used to inhibit the RdRp activity and does not interfere with the entry of SARS-CoV-2. Can CoronaFISH be exploited to detect vRNA in cytosol before its replication?

The aim of our Remdesivir experiment was to test the sensitivity of CoronaFISH, in particular the ability to detect a small fraction of positive cells among a vast majority of negative cells. In principle, CoronaFISH should be applicable to detect incoming vRNA, before replication takes place. However, validating this would require specific experiments and protocol optimization that are beyond the scope of this paper.

R 3.2

N protein usually binds viral RNA genome and I am wondering whether N protein can dramatically decrease the sensitivity of CoronaFISH, especially when it was used to detect the viral particles (for example, samples from environment).

The clear detection of the viral RNA genome by CoronaFISH in infected cells is evidence that N protein binding to the genome does not prevent effective hybridization of the probes in the intracellular environment. We cannot exclude that N protein diminishes the sensitivity of CoronaFISH in other conditions such as environmental samples, but testing this is outside the scope of our paper. We note, however, that because our probe set covers the entire viral genome, CoronaFISH should be less susceptible to diminished sensitivity than other hybridization approaches that only target specific regions of the viral RNA.

R 3.3

The authors would better compare the sensitivity of CoronaFISH with IF (anti N protein or S protein) to support their claim.

We validated CoronaFISH detection using the J2 antibody to visualize dsRNA (see response to R1.2 and Fig 1f).

R 3.4

The detection limit of CoronaFISH for the samples from nasal swabs should be provided and compared with other methods such as QPCR and CRISPR.

Fully addressing this would require lengthy experiments on a large number of additional samples, which we consider to be outside the scope of this paper.

R 3.5

In the inhibition assay, the reproducibility of CoronaFISH should be provided. And the IC50 detected by CoronaFISH could be compared with other methods.

We actually did not measure IC50 by CoronaFISH. However, this comment prompted us to measure the IC50 more carefully using the focal forming assay, by varying Remdesivir concentrations from 0.3 uM to 10 uM. Qualitative inspection and quantitative analysis of the CoronaFISH images confirms a reduction in the number of infected cells. Interestingly, at the highest concentration of 10 uM, the viral output is reduced by 100-fold relative to untreated cells, but a small fraction of cells is still infected. Using CoronaFISH, we could also detect some rare infected cells at this high drug concentration, further validating the sensitivity of our technique. We included these results in Fig 2, panels e-f and Fig S2, panels d-e.

R 3.6

More data in detecting replication kinetics by CoronaFISH should be provided to support the author's statement. Have the authors tried to detect the entry and replication process of SARS-Cov-2 RNA by CoronaFISH with a time span?

A thorough analysis of viral entry and/or replication kinetics is beyond the scope of this methods oriented paper. However our paper actually includes CoronaFISH data at multiple time points after infection (6h p.i.: Fig. 1f; 18h p.i.: Fig. 1d,g, S1a; 24 h p.i.: Fig. 2b,f; 29 h p.i.: Fig S6; 36 h p.i.: Fig. 3a,b) and for all these time points we observe clear signal. Together with our new analyses of Remdesivir treated cells (see previous comment), these data further demonstrate the sensitivity of CoronaFISH and its potential for future analyses of replication kinetics.

Minor points

R 3.7 , line 241, "samples will be required to assess specificity and specificity" should be " samples will be required to assess specificity".

We thank the reviewer for pointing out this repetition, which we deleted.

R 3.8. Figure 4, b-f should be shown in sequential order in the paper.

Reordering these panels would artificially increase the size of the figure and create unnecessary white space. We will leave it to the editor to decide whether and how to reorganize this figure.

R 3.9. Did all 96 probes be verified individually?

A single probe does not provide a sufficiently strong signal to be distinguished from a non-specifically bound probe. This is why smFISH methods use a pooled set of oligos to obtain a strong enriched and specific signal. Individual probes are only tested when a pool of oligos gives strong non-specific background, which is not the case for our probes.

November 15, 2021

RE: Life Science Alliance Manuscript #LSA-2021-01124-TR

Dr. Christophe Zimmer
Institut Pasteur
Unité Imagerie et Modélisation
25, rue du Docteur Roux
Paris 75015
France

Dear Dr. Zimmer,

Thank you for submitting your revised manuscript entitled "Sensitive visualization of SARS-CoV-2 RNA with CoronaFISH". We would be happy to publish your paper in Life Science Alliance pending final revisions necessary to meet our formatting guidelines.

- please upload your main and supplementary figures as single files;
- please add ORCID ID for the corresponding (and secondary corresponding) author-both of you should have received instructions on how to do so
- please add a Summary Blurb/Alternate Abstract in our system
- please add the Twitter handle of your host institute/organization as well as your own or/and one of the authors in our system
- please consult our manuscript preparation guidelines <https://www.life-science-alliance.org/manuscript-prep> and make sure your manuscript sections are in the correct order
- please add an Author Contributions section to your main manuscript text
- please use capital letters when introducing panels in the figures, their legends, and callouts in the manuscript text
- please add your main and supplementary figure legends to the main manuscript text after the references section
- there are callouts for figure 4D-F in the manuscript text, please revise
- please add callouts for Figures 5A-F and S2D to your main manuscript text

A. FINAL FILES:

B. MANUSCRIPT ORGANIZATION AND FORMATTING:

Sincerely,

December 22, 2021

RE: Life Science Alliance Manuscript #LSA-2021-01124-TRR

Dr. Christophe Zimmer
Institut Pasteur
Unité Imagerie et Modélisation
25, rue du Docteur Roux
Paris 75015
France

Dear Dr. Zimmer,

Thank you for submitting your Research Article entitled "Sensitive visualization of SARS-CoV-2 RNA with CoronaFISH". It is a pleasure to let you know that your manuscript is now accepted for publication in Life Science Alliance. Congratulations on this interesting work.

DISTRIBUTION OF MATERIALS:

Again, congratulations on a very nice paper. I hope you found the review process to be constructive and are pleased with how the manuscript was handled editorially. We look forward to future exciting submissions from your lab.

Sincerely,
